# The Socio-Economic Impact of Migration on the Labor Market in the Romanian Danube Region

**Bogdan-Cristian Chiripuci** [1,*], **Marius Constantin** [1,*] , **Maria-Floriana Popescu** [2,*] and **Albert Scrieciu** [3]

1   Faculty of Agri-Food and Environmental Economics, The Bucharest University of Economic Studies, 010374 Bucharest, Romania

2   Faculty of International Business and Economics, The Bucharest University of Economic Studies, 010374 Bucharest, Romania

3   National Institute of Marine Geology and Geoecology (GeoEcoMar), 024053 Bucharest, Romania; albert.scrieciu@geoecomar.ro

\*   Correspondence: bogdan.chiripuci@gmail.com (B.-C.C.); constantinmarius15@stud.ase.ro (M.C.); maria.popescu@rei.ase.ro (M.-F.P.)

**Abstract:** The dynamics of the labor market have been significantly influenced and impacted in recent decades by the scale of globalization, not only from a socio-economic standpoint, but also from the perspective of generating the premises of destroying boundaries. Taking this into consideration, our research is aimed at highlighting the socio-economic impact of migration on the labor market in the Romanian Danube Region in order to create a framework which can be used when elaborating a solid action plan meant to increase the socio-economic attractivity in the analyzed region. This research was carried out by resorting to the multicriterial analysis, aimed at quantifying the state of development of the analyzed counties from the perspective of clearly defined socio-economic indicators. Additionally, the linear regression research method was applied in order to ensure a more in-depth analysis of the relationship between local employment and the departure from domicile. The main finding was that Mehedinți County has greater potential for socio-economic development than the neighboring county, even though the population is not encouraged to remain in the rural areas, one of the reasons being the lack of attractiveness of the local labor market. The designed econometric model confirms (66.17%) this result in the case of the localities part of Mehedinți County.

**Keywords:** labor market; employment; migration; Danube Region; Romania

## 1. Introduction

In recent decades, the process of globalization has dramatically increased international migration, especially from poor to rich countries. A highly significant topic on the foreign research and policy agenda is the analysis on migrants and their economic motives, alongside the impact this flow has on the labor market from both countries (the donor and the receiver). The growing percentage of aging population and the increasing demand for personal and family service in rich countries have increased the demand for foreign workers who are able to fill these gaps. Moreover, highly educated staff, in particular scientists and engineers, have increased their international mobility, creating an international competition for talent, needed to drive innovation in research and technology in advanced sectors, drivers and pinpoints of the current era. Migration was thus a phenomenon of considerable significance in expanding opportunities and maintaining socio-economic transition in advanced market economies at the beginning of the twenty-first century, a significant role being played by both professional and unskilled immigrants.

When talking about such issues as the ones mentioned above, we cannot overlook the area of the Danube, which is one of Europe's most vulnerable regions, with a population in decline [1]. The population is forecasted to decline in the region [2], which will eventually affect the planned economic growth, market shifts, social preferences, etc. On one side, not only the same river and area, but also the same problems are shared by the 19 Danube countries. On the other hand, the region's composition is very heterogeneous, as it is one of Europe's most complex and mosaic-modeled territories ethnically, linguistically, religiously and culturally [3]. Therefore, an adequate level of support is vital to ensure the cross-sectoral policy alignment between appropriate government agencies, educational authorities, social and health services, cultural policies authorities, housing and urban planning bodies as well as asylum and immigration services at the local, regional and national levels, alongside with the ensuring dialog with civil society in the Danube Region. Ensuring appropriate consideration for the rural areas in the Danube Region is vital, especially considering that it has been demonstrated that inhabitants of the Romanian rural areas are more attracted to other rural areas, from the United Kingdom for example [4], than to the Romanian rural areas.

The Danube River Basin (DRB) is the largest river basin in the European Union and occupies an area of 801,463 km$^2$ [5]. It is shared between 19 nations, making it the most foreign river basin in the world [6]. The Danube River Basin is distinguished not only by its scale and large number of countries, but also by its complex ecosystems and significant socio-economic disparities [7]. The Danube River Basin has an amazing variety of ecosystems, including high-gradient glaciated mountains, forested midland mountains and hills, upland plateaus, plains and wetland lowlands (i.e., the Danube Delta, near sea level) [8]. Managing the Danube River Basin has a high degree of difficulty, given the ecological and socio-economic characteristics of the system, and the many issues associated with it, such as surface and water pollution, on-going transformation and erosion, etc.

In 1992, the Rhine–Main–Danube canal was opened [9], and the direct connection was formed between the Danube River and the entire waterway network in Western Europe. This assures a great transcontinental link between Western Europe and the Black Sea and the Danube Area and the harbors of Antwerp, Rotterdam and Amsterdam, which has always and continues to carry great expectations.

A community of 19 countries, with a heterogeneous economy—wealthy western (Austria, Germany), very well developed central (Hungary, Czechia, Slovakia and Slovenia) and poorer southern and eastern (Bosnia and Herzegovina, Bulgaria, Croatia, Montenegro, Romania and Serbia)—is part of Danube Region. This heterogeneity becomes even greater when the countries are divided into regions, as the economic development disparities between the regions are even more prominent than the national differences. A total of 65 areas can be divided within Danube Area, which corresponds to the EU's Nomenclature of Territorial Units for Statistics 2 division, which splits basic regions into groups for the application of regional policies [10].

In fact, although closely interconnected (culturally and socially), the Danube Region is very economically and politically heterogeneous [11]. There are very large differences in the area, as stated in the European Commission's document: "The Region encompasses the extremes of the EU in economic and social terms. From its most competitive to its poorest regions, from the most highly skilled to the least educated, and from the highest to the lowest standard of living, the differences are striking. The Strategy reinforces Europe 2020, offering the opportunity to match the capital-rich with the labor-rich, and the technologically-advanced with the waiting markets, in particular through expanding the knowledge society and with a determined approach to inclusion. Marginalized communities (especially Roma) in particular should benefit. [ . . . ] Roma communities, 80% of whom live in the Region, suffer especially from social and economic exclusion, spatial segregation and sub-standard living conditions." [12].

The transboundary management of the river basin in the Danube has always been extremely important given the number of countries and the complexity of social, political and economic conditions. The convention regarding the safety of the Danube River, which led to the establishment of the International Commission to Protect the Danube River (ICPDR), was signed in 1994. The ICPDR

serves as a forum for the implementation in the Danube River Basin district of the EU Water Framework Directive (adopted in 2000 under the name "Directive 2000/60/EC of the European Parliament and of the Council establishing a framework for the Community action in the field of water policy") and the EU Floods Directive (it was adopted in 2007 under the name "Directive 2007/60/EC on the assessment and management of flood risks"). As an organizational mechanism for water quality monitoring in the Danube River Basin, the ICPDR has established a transnational monitoring network and developed a range of permanent expert bodies proactively dealing with issues such as flood control risk management, surface water monitoring and evaluation, groundwater and other related topics. The network of experts has been further expanded to cooperate with other international organizations and institutions that led to an increase in the amount of data obtained over time, with a better quality and accuracy.

Moreover, the European Strategy for the Danube Region (EUSDR) has also experienced a turbulent evolutionary journey since its launch in 2011. It is not possible to grasp how well integrated EUSDR has been and still is in the larger EU and its latest New Member States, needing a broader perspective [13]. The EUSDR is therefore inseparably connected to larger globalization processes and can only be seen as a relatively short chapter in the very complex history of the European Union. The area under the EUSDR includes 115 million people [14], from the Black Forest (Germany) to the Black Sea (Romania–Ukraine–Moldova).

The scientific literature on the field of the Danube Region is not very wide in terms of articles or books, but there are some topics related to the DRB that need to be mentioned: environmental issues [15–18], socio-economic issues [19–21], strategic and political issues [13,22–28] or country-related articles (seven countries that have a percentage of the DRB more than 5% were chosen—Austria (10%) [29–32], Bulgaria (5.9%) [33–35], Germany (7%) [36–38], Hungary (11.6%) [35,39–42], Romania (29%) [15,34,43–46], Serbia (10.2%) [20,47–50] and Slovakia (5.9%) [51].

Therefore, the study of the management of the water basin is always needed due to the lack of local expertise, high administrative and socio-economic complexity, the diverse interests of stakeholders and difficulties in implementing international and domestic law [52].

One of the objectives undertook in this research paper was to elaborate a framework of premises aimed at consolidating an action plan meant to increase the socio-economic attractivity of the Dolj and Mehedinţi Counties, which are situated in the vicinity of the Danube River. In order to do so, a multicriterial analysis was carried out. Moreover, other aim of this research which supports the previously mentioned objective was to study the relationship between the average number of employees and the number of departures from the domicile, at the level of the localities part of the Dolj and Mehedinţi Counties. This analysis allows one to have a better perspective on migration, while also studying the dependencies between the local employment situation in the Danube Region and the lack of attractiveness in this region (analyzed from the perspective of departures from the domicile). Starting with a multicriteria analysis meant to evaluate important characteristics of Mehedinţi and Dolj Counties, this study goes deeper into the labor market analysis, from the county level analysis (multicriteria method) to the localities level analysis (econometric method, based on the initial findings of the multicriteria analysis), aiming to offer a deep analysis of the socio-economic potential of the area and pursuing the provision of various solutions aimed at the development of the area. Hence, our paper aims to tackle the issues of migration and the labor market into the socio-economic environment in the Danube Region in Romania. The more the Danube Region represents a territory with decreasing barriers, the more the risk of population concentration in large metropolitan agglomerations with higher socio-economic standards will increase.

## 2. Materials and Methods

In order to carry out a relevant analysis of the rural area of the Mehedinţi and Dolj Counties, we present a case study, which aimed to build a framework that contained all the premises for future research in order to achieve a management plan for the development of this area. The two counties

were chosen for analysis, because they represent the wettest area in Romania, crossed by the Danube, except for Tulcea County, where this river creates the Danube Delta. In this regard, a continuity of the socio-economic perspective of the rural area in this southwestern part of Romania will be pursued. In order to design a framework for the sustainable development of villages in the vicinity of the Danube, an assessment of the potential of the area, including at the level of human and natural resources, is necessary. Thus, aspects of population migration were addressed in an econometric manner to see if there were differences between this indicator at the level of the two counties analyzed. In the domain of sustainable development, the need for the current improvement of river infrastructure, the development of agritourism, agriculture and the creation of recreational activities based on the use of lakes, ponds, wetlands or the stretch of a small river located in the rural area of Mehedinți and Dolj Counties was pursued. The Danube, representing a huge collection basin, can offer various solutions for navigation, hydropower plants, fish farming, being able to provide water for agriculture, industry and population. All these aspects must be incorporated at the level of regional policies on the development of globalization that take into account socio-economic aspects at the level of rural space and customized according to the specifics of the area [53–56].

In this regard, it was decided to use the method of multicriterial analysis, which the authors called the determination of the socio-economic situation of an area in the immediate vicinity of the localities on the Danube, from Romania. This method is used to quantify socio-economic transformation according to its own parameters of expression (own parameters of behavior), expressed on the basis of a series of clearly defined economic indicators and techniques, doubled by information and qualitative data [57].

This quantification model requires several steps, as follows:

- Making the list of indicators, on the basis of which can be identified the perspective of determining the socio-economic situation of the Mehedinți–Dolj area, located near the Danube and assigning coefficients of importance for each indicator.
- The values of the coefficients must be between 5 and 10, depending on the influence of the potential of the rural area analyzed (for 5–6, the degree of importance is reported as a secondary one, 7–8 is considered as major, and 9–10 is considered as of great importance).
- The determination of the rank for the 10 indicators at the level of the two counties located in southwestern Romania, Mehedinți and Dolj (setting the rank of each county according to the reference area; for establishing the hierarchy, the grades awarded will be 1 and 2, including the average of each county). This indicator will determine the setting of priorities at the criterion level, which is particularly important in any analysis study.
- Calculation of the aggregate note at the level of each indicator according to Table A1 (Appendix A):

$$Nagik = Rk \times C \tag{1}$$

where:

- i = indicator (1–2);
- K = county (Mehedinți, Dolj).

Identification of the global indicator for each county. This step includes the summation of the aggregate notes resulting from each analyzed county:

$$Ik = \sum\nolimits_{i=1}^{2} Nagi \tag{2}$$

In order to identify the socio-economic perspective of the rural area, located in the vicinity of the Danube, the method of multicriteria analysis was used at the county level, to establish differences between the two areas and the potential they hold. In applying this method, 10 indicators were used, which are considered driving factors in determining the socio-economic situation of the rural area in Mehedinți and Dolj Counties.

The 10 indicators in Table 1 received a value based on the importance given in determining the socio-economic situation of the Mehedinți–Dolj rural area, which is located near the Danube. In this regard, the values attributed to the chosen indicators were distributed according to the information provided by the mentioned bibliography references and the authors' vision of the development of the labor market from the two analyzed counties. In this way, a harmonious combination is created between the theoretical and practical part of the research, which aims to provide results of the analysis, as relevant as possible on the subject approached. Therefore, the lowest value was attributed to the indicator "Creating prospects for accessing structural funds in rural areas, within the framework of the measures specific to non-agricultural activities", which encompasses all non-agricultural economic activities. In order to separate the possibilities of developing the area near the lakes, ponds, wetlands or stretch of a small river flowing into the Danube, an independent indicator was assigned, namely "Recreational activities in the rural area". It will incorporate both the perspectives of existing and future entrepreneurs on innovative ideas that must generate new jobs in the area under review. At the same time, there is the possibility of developing the county to a different area of activity than the areas aiming to use the Danube or wetland in the countryside. For example, Dolj is the county where Craiova has attracted a large foreign investment in Oltenia, to the detriment of counties such as Gorj or Mehedinți. However, after this success, the local administration, in order to achieve the sustainable development of the area, tried to establish a metropolitan area, a disadvantaged area, and multiplied the number of municipalities, cities and communes. However, these decisions did not have the expected results, failing to stimulate entrepreneurial initiatives [58]. This indicates that this decision at the time was not a winning one, but in the future it could help to stimulate public–private partnerships or the chain development of various economic activities, closely linked, including the river area of the counties analyzed.

Regarding the indicators with the highest values, we list "Development of infrastructure on the basis of the Danube River", "Creating new jobs through economic activities carried out in the vicinity of the Danube" and "Repopulation of villages and communes near the Danube". The last two indicators define the result of econometric regression, while the first indicator targets a broad entrepreneurial perspective, equally important for the repopulation of rural space by creating new jobs in economic activities that take place in the vicinity of the Danube. Subsequently, values between 7–8 were allocated to the rest of the coefficients of importance, in order to achieve a dynamic on the socio-economic perspective at county level. This allocation of rank was carried out with the aim of building a ranking of the two southwestern counties of Romania, based on the need to develop the rural area in the vicinity of the Danube. The value of the indicators was also reported according to the importance of each on the current perspectives of the rural environment in the area under review, according to the views of some experts in the field, found in [59–62].

This analysis will allow us to highlight both the needs and perspectives that the rural areas of Mehedinți and Dolj counties possesses. Additionally, following this case study, we can provide a relevant framework on the results that can be the basis for the realization of a management plan at the local and regional level regarding the development of the analyzed area, located in the vicinity of the Danube.

Moreover, to continue our research, we chose an analysis based on linear regression, therefore a quantitative research method, implying an econometric approach. Econometrics is a form of knowledge which includes techniques and methods for analyzing the dynamics of the variables in many fields of activity, as well as the interconnections among variables [63]. The linear regression model offers the possibility to study and confirm the existence or nonexistence of correlations between two types of variables: dependent and independent variables. Considering the objective undertaken to study the relationship between the average number of employees and the number of departures from the domicile, the cross-sectional analysis is suitable in this case. This type of linear regression, based on cross-sectional data, involves the advantage that the analysis is focused at a period closest to a selected moment in time and highlights correlations among the observations [64], in this case: the

Romanian localities part of Dolj County and Mehedinți County. The extracted cross-sectional data are characterized by multiple observations studied at a certain moment in time (in this case, the year 2018), referring to several entities (in this case, the analyzed localities), focusing on a single phenomenon (in this case, both indicators included in this study). The year 2018 was selected as a point of reference, because this is the latest comprehensive statistical data available regarding one of the indicators included in the analysis: the average number of employees. Even though there are more recent data available regarding the number of departures from the domicile, the cross-sectional analysis carried out in this paper is referring to the year 2018, since this is the year statistically compatible from the perspective of both analyzed indicators. What makes this research method unique is the fact that the variables are analyzed considering the same specific period in time that the method is focused on the subjects (also called observations) approached, rather than focusing on how the values associated with the variables change over longer periods of time.

**Table 1.** The importance coefficients for the analyzed indicators.

| Item no. | Indicators | Coefficient of Importance |
|:---:|:---:|:---:|
| 1 | Social perspective from the county countryside | 8 |
| 2 | Economic perspective in the county countryside | 8 |
| 3 | Developing the social relationship within the rural community | 7 |
| 4 | Development of infrastructure on the basis of the Danube River | 10 |
| 5 | Creating new jobs through economic activities carried out in the vicinity of the Danube | 10 |
| 6 | Development of agritourism | 8 |
| 7 | Creating prospects for access to structural funds in rural areas, within the framework of measures specific to agricultural activities | 7 |
| 8 | Creating prospects for access to structural funds in rural areas, within the framework of measures specific to non-agricultural activities | 6 |
| 9 | Recreational activities in the countryside | 8 |
| 10 | Repopulation of villages and communes in the vicinity of the Danube | 10 |

Source: authors' own conceptualization.

Data were taken from the databases of the Romanian National Institute of Statistics. The area covered is that of the 61 localities part of Mehedinți County and 104 localities part of Dolj County. In the cases of both counties, municipalities and towns were excluded from the observations. The cross-sectional linear regression methodology was applied on the available data using the least-squares method. This is a common, standard approach in analyses specific to regressions, used to approximate the solution by minimizing the sum of the squares of the residuals (also named errors) made in the results of every single equation. Results that fit best in the model are those which minimize the sum of squared residuals/errors.

The econometric model was designed to study in structure the relationship between the dependent (also called endogenous) and the independent variable (also called exogenous variable). In this econometric model, the endogenous variable is the number of departures from the domicile and

the exogenous variable is the average number of employees by localities. These indicators have a code assigned by Romanian National Institute of Statistics, as it follows: POP308A in the case of the number of departures from the domicile (including external migration) and FOM104D in the case of the average number of employees by localities. According to the Romanian National Institute of Statistics, the change of domicile is registered only if the persons who left a given locality, proved to have ensured a dwelling in another different locality, while taking into account that changes of domicile from one street to another within the same locality and from on village to another within the same commune were not included in the statistics. These data include international emigrants and the number and name of localities are in accordance with administrative territorial structure updated for current year. Regarding the exogenous variable, the Romanian National Institute of Statistics states that the average number of employees comprises all persons with an individual labor contract or agreement for either a definite or indefinite duration, while also including seasonal workers, the manager or the administrator, with the restriction that the labor contract or agreement must not have been suspended during the reference year. The average number of employees represents a simple arithmetic mean resulting from summing up the daily number of employees (considering that the suspended labor contracts and agreements were excluded), including weekends, legal holidays or other non-working days, divided to the total number of calendar days. The daily number of employees taken into account for the compilation of the average number of employees comprises all persons with an individual labor contract or agreement for a definite or indefinite period of time, while also including seasonal workers, the manager or the administrator, whose labor contract or agreement must not have been suspended during the reference year. Based on these indicators, an initial descriptive statistical analysis was carried out, while not only taking into consideration the permanent resident population (indicator code: POP107D), but also the variable ratios when reported to the permanent resident population. The ratios were calculated by the authors in Appendix B, based on the data extracted from the Romanian National Institute of Statistics platform (TEMPO Online).

## 3. Results

Starting with the multicriteria analysis, the overall indicator was calculated as a sum of the aggregate notes obtained for each county located in southwestern Romania, within the 10 indicators mentioned above. Depending on the value of the importance of each indicator, the global indicator was determined, which led to the discovery of the area in the rural area that benefits from the highest potential for the development of economic activities, which can be carried out in the vicinity of the Danube, in the territory of Romania. As can be seen in Table 2, the highest level of the global indicator was found in Mehedinți County, which indicates that this area has a greater potential than the neighboring county on the prospect of socio-economic development at the level of the rural area, adjacent to the river which holds the second length at European level. Dolj County is located in a predominant area of the plain, where the main investment in the infrastructure related to the Danube would be in irrigation, necessary for the agricultural area, as well as in the development of recreational areas (restoring parts of the former floodplain), close to the various river basins. On the other side, Mehedinți County presents in addition, hilly and even mountainous areas, which gives the relief an interesting dynamic, and allows it to enjoy a diversity of economic activities.

**Table 2.** Determination of the global indicator.

| County | I1 | I2 | I3 | I4 | I5 | I6 | I7 | I8 | I9 | I10 | TOTAL |
|--------|----|----|----|----|----|----|----|----|----|-----|-------|
| **Mehedinți** | 16 | 16 | 14 | 10 | 10 | 8 | 7 | 12 | 16 | 20 | 129 |
| **Dolj** | 8 | 8 | 7 | 20 | 20 | 16 | 14 | 6 | 8 | 10 | 117 |

Source: authors' own conceptualization, based on the multicriteria analysis method.

The values assigned to Table 2 are calculated according to the method of performing the multicriteria analysis (more precisely the determination of the global indicator) mentioned above and the value of the importance coefficients mentioned in Table 1. In this way, the determination of the global indicator is closely related to the previously reported information. In addition, regarding the analyzed area, from southwestern Romania, it was found that the Danube reaches here for the first time on Romanian soil and then travels a distance of 354.1 km on the right bank and 1050 km on the left bank, to the delta river, in Tulcea County [65]. Based on the existence of the Danube in this area, the hydropower plants Iron Gates 1 and 2 were created in Mehedinți County, some of the largest such hydrotechnical constructions in Europe and the largest on the Danube [66]. At the same time, this county includes municipalities such as Drobeta-Turnu Severin, Orșova, Baia de Aramă, Strehaia and Vânju Mare, which run on an area of 4933 km$^2$, with a population density of 52 inhabitants/km$^2$, according to the data centralized in the 2011 Census [67]. All these localities have under their subordination different communes, which contain lakes, ponds, wetlands or stretches of a small river that later flow into the Danube. According to the econometric regression model, there is a preference for migration in the case of localities with high levels of population (when reported to the county level), which should be treated by various local measures and policies. The problem exists at the local level, because when considering the regional level, there is another preference when considering population migration in Dolj County. This preference is specific to a constant number of departures from the localities, but it still allows entrepreneurs to constantly employ staff.

According to Table 3, the average number of employees by locality is characterized by a mean of 150 employees in the case of the localities part of Dolj County, with a standard deviation of almost 246 employees. Since the standard deviation is almost twice the mean, it is obvious that there are serious development discrepancies between the analyzed localities, especially from the perspective of employment opportunities. While job security is a major concern in some localities (for example, the locality with only 27 employees), other localities are exceptions, reaching the maximum value of 1601 average employees (a locality named Ișalnița, with a total population of 4045).

**Table 3.** Descriptive statistics regarding the analyzed indicators (cross-section), reference point: the year 2018, localities part of Dolj County.

| | DOLJ | |
| | FOM104D | POP308A |
|---|---|---|
| Mean | 150.1538 | 50.9327 |
| Median | 77 | 49 |
| Maximum | 1601 | 143 |
| Minimum | 27 | 7 |
| Standard deviation | 245.6192 | 26.2147 |
| Skewness | 4.3580 | 0.8687 |
| Kurtosis | 23.9619 | 3.7199 |
| Jarque–Berra | 2233.2930 | 15.3267 |
| Observations | 104 | 104 |

Source: authors' own calculations (data source: The Romanian National Institute of Statistics, processed in EViews 10 Student Version Lite, IHS Global Inc., Irvine, CA, USA).

The distribution of the average number of employees by localities highlights the discrepancies between the analyzed localities even more. In Figure 1, Skewness reflects the positive asymmetric distribution of data around the mean [68], considering that the value is above zero (4.3580 to be more precise). This indicates that the majority of the localities are at almost the same state when analyzing from the perspective of employment. Additionally, this fact is supported by Kurtosis, an indicator which reflects how flat or curved a distribution is compared to a normal one. A normally distributed series implies a Kurtosis value of three [69]. Considering the value of Kurtosis of almost 24 in the case of this distribution, the leptokurtic characteristic of the distribution is obvious, further increasing the

finding that there are serious development discrepancies between the analyzed localities. However, even though major discrepancies are noticeable, there are not many.

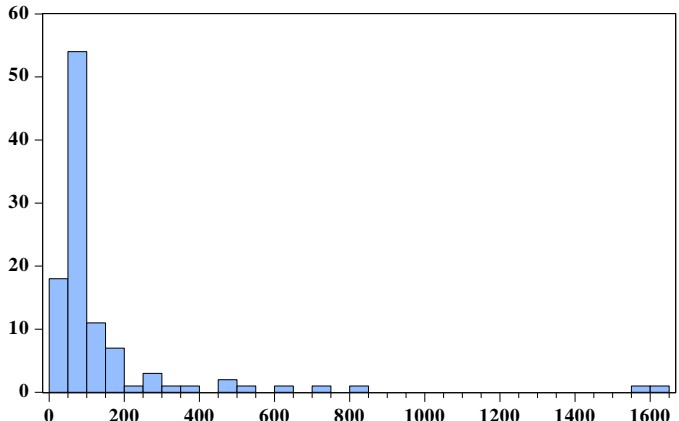

**Figure 1.** The distribution of the average number of employees by localities. Reference point: the year 2018, the case of the localities part of Dolj County Source: authors' own representation (data source: The Romanian National Institute of Statistics, processed in EViews 10 Student Version Lite, IHS Global Inc., Irvine, CA, USA).

The number of departures from the domicile is on average close to 51 in the case of the analyzed localities part of Dolj County, with a standard deviation of 26 departures. While the minimum number of departures is 7, the maximum value is 20.42 times greater than the minimum, signaling discrepancies once more. On the other hand, when comparing the distribution of the number of departures from the domicile to that of the average number of employees by localities, one can notice that the distribution of the number of departures from the domicile is the one that is closer to a normal distribution. In Figure 2, the Skewness value of 0.8687 indicates the tendency towards positive asymmetry and the Kurtosis value of 3.7199 indicates the tendency towards a leptokurtic distribution. Taking these characteristics into consideration, one can notice that there are only few outlier localities when analyzing of the number of departures from the domicile.

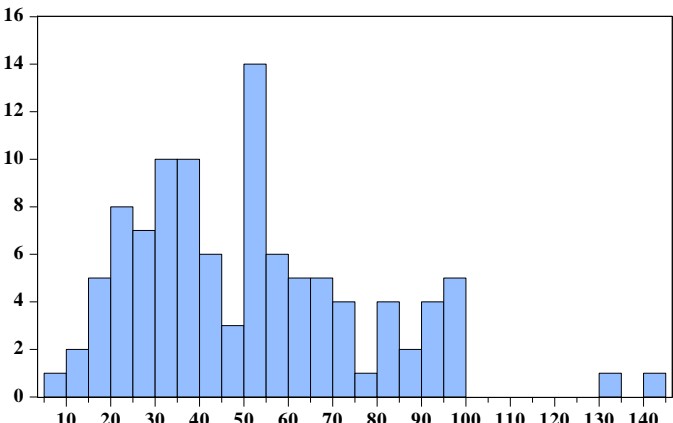

**Figure 2.** The distribution of the number of departures from the domicile. Reference point: the year 2018, the case of the localities part of Dolj County Source: authors' own representation (data source: The Romanian National Institute of Statistics, processed in EViews 10 Student Version Lite, IHS Global Inc., Irvine, CA, USA).

In Figure 3, the scatter plot in the case of the number of departures from the domicile and the average number of employees by localities brings forward the low correlation between the two indicators (34.80%) and that there are few localities within Dolj County that are considered outliers

(spread far from the linear regression line). This results in a possible finding—that the outliers have the potential to become towns if investments are ensured and if migration is tempered.

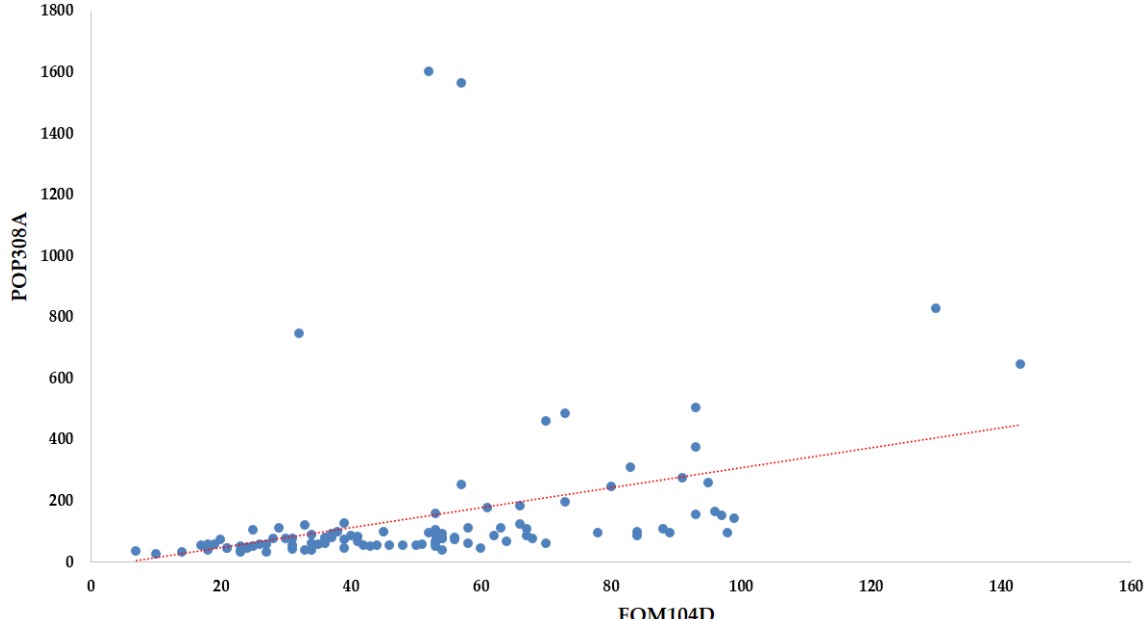

**Figure 3.** The scatter plot in the case of the number of departures from the domicile and the average number of employees by localities. Reference point: 2018, the localities part of Dolj County. Source: authors' own representation (data source: The Romanian National Institute of Statistics, processed in EViews 10 Student Version Lite, IHS Global Inc., Irvine, CA, USA).

According to Table 4, the 61 localities part of Mehedinți County in 2018, the average number of employees by localities is characterized by a mean of almost 134 employees, 16 employees less than in the case of the 104 localities part of Dolj County. Similarly to the situation in Dolj, development discrepancies are visible between the analyzed localities in the case of Mehedinți County from the perspective of employment opportunities, especially when considering the value of the standard deviation, almost 183 employees. While employment is an issue in some localities (see the locality with only 24 employees—which is the minimum), other localities are at a completely different stage of engaging in economic activities (1397 employees, the maximum value registered in the case of a commune named Șimian, situated extremely close to the Danube and with a population of 10.346).

**Table 4.** Descriptive statistics regarding the analyzed indicators at the level of localities (cross-section), reference point: the year 2018, Mehedinți County.

| | MEHEDINȚI | |
| --- | --- | --- |
| | **FOM104D** | **POP308A** |
| Mean | 133.9180 | 49.1639 |
| Median | 81 | 42 |
| Maximum | 1397 | 221 |
| Minimum | 24 | 11 |
| Standard deviation | 182.6559 | 32.0266 |
| Skewness | 5.6264 | 2.7638 |
| Kurtosis | 38.8950 | 14.8490 |
| Jarque–Berra | 3596.6448 | 434.5017 |
| Observations | 61 | 61 |

Source: authors' own calculations (data source: The Romanian National Institute of Statistics, processed in EViews 10 Student Version Lite, IHS Global Inc., Irvine, CA, USA).

Once again, based on Figure 4, the distribution of the average number of employees by localities highlights the discrepancies between the analyzed localities, Mehedinţi being no exception. The Skewness value of 5.6264 indicates a strong positive asymmetry and that most of the observations tend towards the minimum value rather than towards the maximum value. Moreover, taking into account the value of Kurtosis of almost 39 in the case of this distribution, the leptokurtic characteristic is a defining one, furthermore validating the finding that there are very few outliers which have the potential to become towns if measures are being considered in order to increase the attractiveness of the analyzed localities.

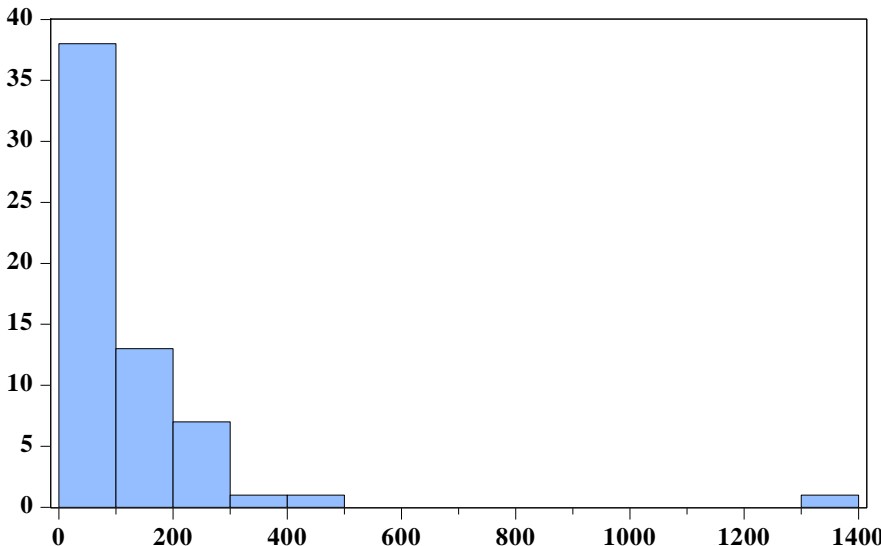

**Figure 4.** The distribution of the average number of employees by localities. reference point: the year 2018, the case of the localities part of Mehedinţi County. Source: authors' own representation (data source: The Romanian National Institute of Statistics, processed in EViews 10 Student Version Lite, IHS Global Inc., Irvine, CA, USA).

Analyzing the number of departures from the domicile, the mean is around 49 departures in the case of the localities part of Mehedinţi County, with a standard deviation of 32 departures. While the minimum number of departures from is 11, the maximum value is 20.09 times greater than the minimum, signaling discrepancies again and possibly the fact that big localities (reported to the county level) are no longer attractive to the population.

Moreover, the data processed in the Appendix B are essential for the descriptive statistical analysis, because they highlight the relevant characteristics of the localities, based on the variable ratios when reported to the permanent resident population. Tables A2 and A3 confirm that there are no major discrepancies when reporting the number of departures to the total resident population per locality. Analyzed at the level of localities, in Dolj, the mean of the number of departures–total resident population ratio is 1.75% with a standard deviation of 0.38%. Similarly, in Mehedinţi, the mean of the number of departures–total resident population ratio is 2.23%, with a standard deviation of 0.63%. On the other hand, the number of employees–total resident population ratios signal discrepancies in the case of Dolj when compared to Mehedinţi. This is because the Carcea, Ghercesti and Isalnita localities registered favorable percentages of the number of employees reported to the total resident population, way above the mean (5.03%): 61.20%, 44.23% and 39.58%. Even though these are not the localities with the most resident population in Gorj County, the local workforce in these particular localities is stable. Unfortunately, the number of employees reported to the total resident population signals the lack of engagement in the local labor market, the lack of entrepreneurial initiatives, or both. According to the Romanian National Institute of Statistics data, in Dolj County, 43.59% of the total resident population

is represented by persons situated in the 0–19 and 60+ years old intervals. The situation is almost identical in Mehedinți County: 42.99%.

Surprisingly, when comparing the distribution of the number of departures from the domicile (Figure 5) to that of the average number of employees by localities (Figure 4), one can notice they are similar from some perspectives. Both distributions are deeply leptokurtic and characterized by positive asymmetry. These characteristics are even more pronounced in the case of the distribution of the average number of employees by localities compared to that of the distribution of the number of departures from the domicile.

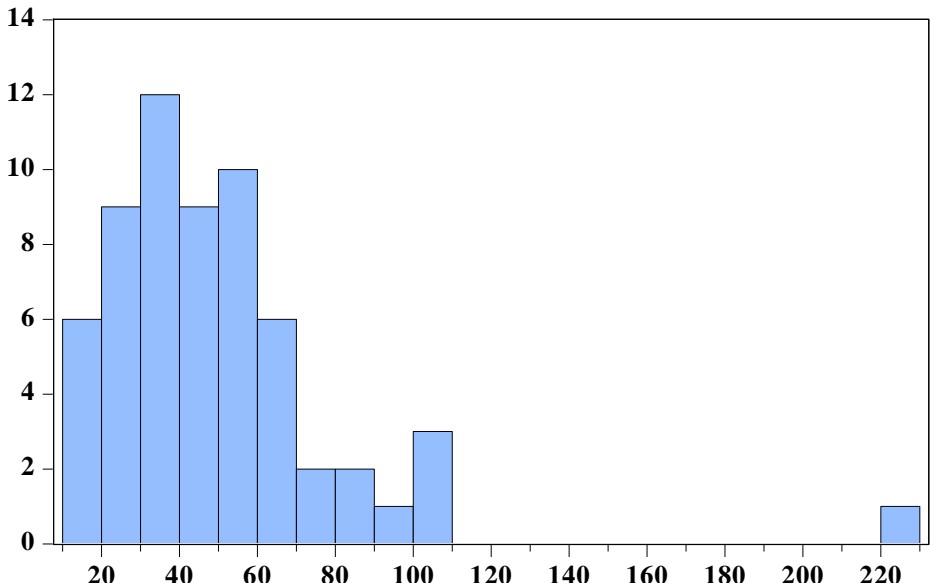

**Figure 5.** The distribution of the number of departures from the domicile. Reference point: the year 2018, the case of the localities part of Mehedinți County. Source: authors' own representation (data source: The Romanian National Institute of Statistics, processed in EViews 10 Student Version Lite, IHS Global Inc., Irvine, CA, USA).

The scatter plot in the case of the number of departures from the domicile and the average number of employees by localities part of Mehedinți County, represented in Figure 6, proves the existence of a positive correlation between the two indicators (81.34%). Unlike the situation of the localities part of Dolj County, this scatter plot fits better the linear regression line in the case of the localities part of Mehedinți County. Scatter plots similar to that illustrated in Figure 6 are specific to the standard linear regression models: the relationships are modeled using linear predictor functions.

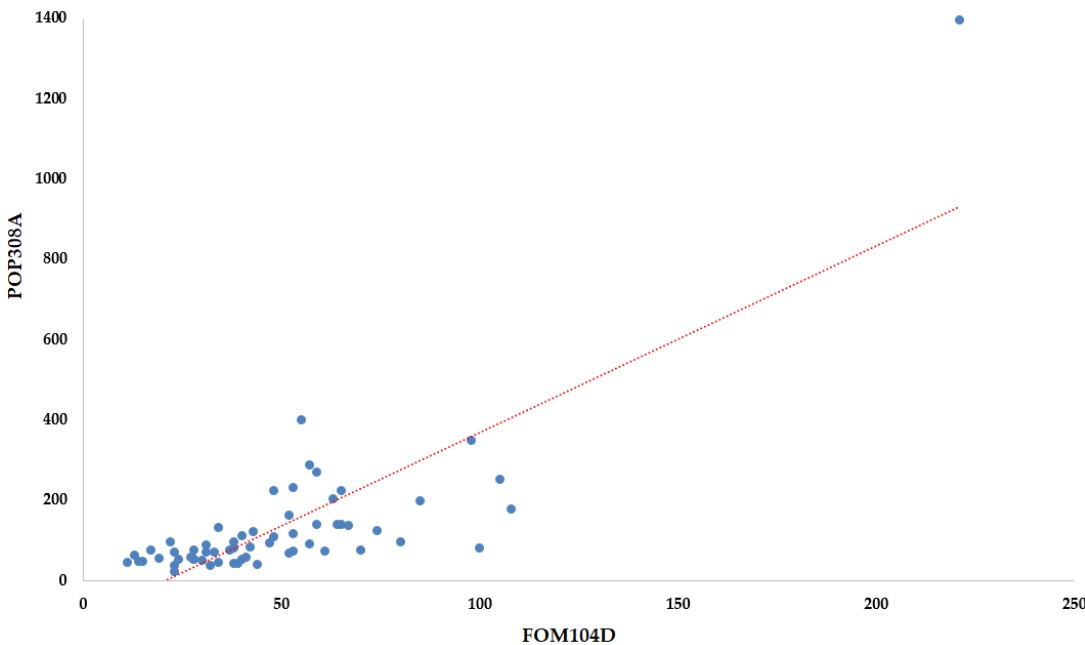

**Figure 6.** The scatter plot in the case of the number of departures from the domicile and the average number of employees by localities. Reference point: 2018, the localities of Mehedinți County. Source: authors' own representation (data source: The Romanian National Institute of Statistics, processed in EViews 10 Student Version Lite, IHS Global Inc., Irvine, CA, USA).

Following the analysis of the descriptive statistics regarding the indicators, the next step in this research was to design the cross-sectional linear regression model. This refers to the relationship between the average number of employees and the number of departures from the domicile, at the level of the localities part of two Romanian counties near the Danube: Dolj and Mehedinți. More specifically, it is meant to provide an equation for the number of departures from the domicile, defined in relation with the average number of employees at the level of the localities at a specific point in time: the year 2018. Table 5 contains more details regarding the designed model.

The coefficient of determination indicates that, only in the case of the localities part of Mehedinți County, 66.17% of the variation of the number of departures from the domicile is explained by the average number of employees. On the opposite side, only 12.11% of the variation of the number of departures from the domicile is explained by the average number of employees in the case of the localities part of Dolj County. From this point forward, the analysis will only focus on the only model that is the closest to successfully defining the dependent variable based on the independent variable—the model referring to the localities part of Mehedinți County. Considering the equation of the beforementioned, should the average number of employees of a locality be situated around the mean in the county, 134 for example, then this triggers a number of departures from the domicile (including external migration) of 49 (calculated: $30.0630 + 0.1426 \times 134$). Therefore, for 70 employees, a locality part of Mehedinți should take into consideration that, according to the designed econometric model, this implies a number of 40 departures from that respective locality.

The Student-t values of the parameters are calculated in the t-Statistic column. If Prob is below 0.05, the null hypothesis is rejected, meaning that the parameters of the variables significantly differ from 0. In the case of this econometric model, the corresponding probability is below 0.05, which results in rejecting the null hypothesis and accepting the alternative hypothesis. The coefficients differ significantly from 0, which validates the constructed model.

In order to counter the mechanical increase in the coefficient of determination [70], Adjusted $R^2$ validates the model, considering that there is a drop of only 0.57 percentage points between the coefficient of determination and the Adjusted $R^2$. The Durbin–Watson statistic is a test for autocorrelation in the

residuals of the model and, in this case, it indicates that successive error terms are slightly negatively correlated, because the value corresponding to this statistic is 2.3408. However, the value is considered acceptable [71].

**Table 5.** The results of the cross-sectional linear regression (least-squares method).

| **Formula of the method** | | | | |
|---|---|---|---|---|
| LS POP308A C FOM104D | | | | |
| **Formula of the equation of the model** | | | | |
| POP308A = C(1) + C(2) × FOM104D | | | | |
| **Equation of the model and coefficients obtained** | | | | |
| POP308A = 30.0630 + 0.1426 × FOM104D | | | | |

**County:** MEHEDINŢI
**Dependent variable:** POP308A
**Method:** least Squares
**Included observations:** 61

| Variable | Coefficient | Std. Error | t-Statistic | Prob. |
|---|---|---|---|---|
| C | 30.0630 | 2.9910 | 10.0513 | 0.0000 |
| FOM104D | 0.1426 | 0.0133 | 10.7430 | 0.0000 |

| | DOLJ | MEHEDINŢI |
|---|---|---|
| $R^2$ | 0.1211 | 0.6617 |
| Adjusted $R^2$ | 0.1125 | 0.6560 |
| S.E. of regression | 231.3873 | 18.7845 |
| Sum squared resid | 5461088.7077 | 20818.6078 |
| Log likelihood | −712.7455 | −264.4535 |
| F-statistic | 14.0601 | 115.4112 |
| Prob(F-statistic) | 0.0003 | 0.0000 |
| Mean dependent var | 150.1538 | 49.1639 |
| S.D. dependent var | 245.6192 | 32.0266 |
| Akaike info criterion | 13.7451 | 8.7362 |
| Schwarz criterion | 13.7960 | 8.8054 |
| Hannan–Quinn criter. | 13.7657 | 8.7633 |
| Durbin–Watson stat | 0.3296 | 2.3408 |

Source: authors' own calculations (data source: The Romanian National Institute of Statistics, processed in EViews 10 Student Version Lite, IHS Global Inc., Irvine, CA, USA).

According to Table 6, the confidence intervals for the variables included in the econometric model confirm the following:

- With a 90% confidence rate: should the average number of employees from any locality part of the Mehedinţi County be 150, then it is estimated that that the respective locality has a corresponding number of departures from the domicile situated in the following interval: 43.1249 lower bound (25.0649 + (150 × 0.1204)) and 59.7812 upper bound (35.0612 + (150 × 0.1648)).
- With a 95% confidence rate: should the average number of employees from any locality part of Mehedinţi County be 150, then it is estimated that that respective locality has a corresponding number of departures from the domicile situated in the following interval: 41.4932 lower bound (24.0782 + (150 × 0.1161)) and 61.4279 upper bound (36.0479 + (150 × 0.1692)).
- With a 99% confidence rate: should the average number of employees from any locality part of Mehedinţi County be 150, then it is estimated that that respective locality has a corresponding number of departures from the domicile situated in the following interval: 38.1968 lower bound (22.1018 + (150 × 0.1073)) and 64.7242 upper bound (38.0242 + (150 × 0.1780)).

**Table 6.** The confidence intervals for the econometric model designed in the case of localities part of Mehedinți County.

| Variable | Coefficient | Low | 90% Confidence High | 95% Confidence Low | High | 99% Confidence Low | High |
|---|---|---|---|---|---|---|---|
| C | 30.0630 | 25.0649 | 35.0612 | 24.0782 | 36.0479 | 22.1018 | 38.0242 |
| FOM104D | 0.1426 | 0.1204 | 0.1648 | 0.1161 | 0.1692 | 0.1073 | 0.1780 |

Source: authors' own calculations (data source: The Romanian National Institute of Statistics, processed in EViews 10 Student Version Lite, IHS Global Inc., Irvine, CA, USA).

In order to have a more in-depth approach of the observations within the model and continue validating it, the residuals were checked via performing the White Test for heteroskedasticity, with the null hypothesis for homoskedasticity.

According to the White Test results included in Table 7, we rejected the null hypothesis and accept homoskedasticity, taking into account that the p-value is above 0.05 threshold. Therefore, we accept that the variance of the residuals is constant and do not vary much as the value of the predictor variable changes. This result validates the designed econometric model.

**Table 7.** Testing the residuals in order to validate the model.

| The White Test | | | |
|---|---|---|---|
| F-statistic | 0.9369 | Prob. F (2,25) | 0.3977 |
| Obs × R-squared | 1.9091 | Prob. Chi-square (2) | 0.3850 |
| Scaled explained SS | 2.7505 | Prob. Chi-square (2) | 0.2528 |

Source: authors' own calculations (data source: The Romanian National Institute of Statistics, processed in EViews 10 Student Version Lite, IHS Global Inc., Irvine, CA, USA).

The residual distribution is normal, as observed in Figure 7, but with some amendments. The mean of the residuals is zero, which is a desirable situation for a model to be valid. However, there is a small tendency towards positive asymmetry (due to the Skewness value of 1.0060, above the ideal zero threshold for a normal distribution), but we consider it acceptable. Kurtosis (4.0800) indicates a leptokurtic distribution of the residuals, not typical for a normal distribution. A more detailed representation of the residuals, per observation, is available in Appendix C.

The perspective of socio-economic development at the level of the areas of southwestern Romania in the vicinity of the Danube, represents real challenges for the local authorities of Mehedinți and Dolj counties, because with the exception of Tulcea County (a fact due to the presence of the Danube Delta), there is no sustainable approach in Romania for the development of non-agricultural activities and implicitly, the repopulation of the related rural area. The advantage of benefiting from a large area of the Danube, in the vicinity of villages and communes, must be seen as a huge potential, still untapped, which can bring the young generation back to the countryside and represent an example of good practice at the national level.

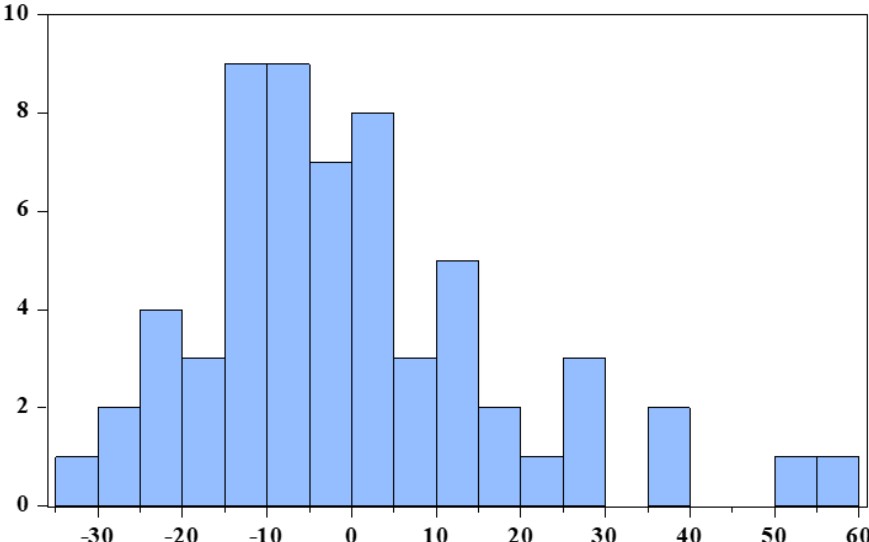

**Figure 7.** The distribution of the residuals. Source: authors' own representation (data source: The Romanian National Institute of Statistics, processed in EViews 10 Student Version Lite, IHS Global Inc., Irvine, CA, USA).

## 4. Discussion

Regarding the first research method, depending on the importance of the indicator, Mehedinți County obtained maximum values when applying the multicriterial analysis methodology within indicators such as: social perspective in the county countryside, economic perspective in the rural county area, the development of the social relationship within the rural community, the creation of perspectives on access to structural funds in rural areas, within the framework of measures specific to non-agricultural activities, carrying out recreational activities in the rural area and the repopulation of villages and communes in close proximity to the Danube. The importance of these indicators at the county level is meant to develop the local socio-economic perspective, so that on the basis of a sustainable social relationship, the migration of the population is diminished. Subsequently, on the basis of the development of the entrepreneurial environment by accessing structural funds in this rural area, will open up new opportunities for jobs in different fields will be available, allowing the repopulation of villages and communes in the vicinity of the Danube.

On the other side, Dolj County encompasses municipalities such as Băilești, Calafat, Craiova, Bechet, Dăbuleni, Filiași and Segarcea, on an area of 7414 km$^2$, with a population density of 89 inhabitants/km$^2$, according to the data centralized in the 2011 Census [67]. All these localities have under their subordination different communes, which contain lakes, ponds, wetlands or the stretch of a small river that later flows into the Danube. Additionally, the town of Bechet is the only point that reaches the bank of the Danube, between the area of the counties analyzed, being an extreme point of the river basin of Romania. At the same time, it can be observed that the density level between the two counties analyzed was different even about 10 years ago, indicating that the migration trend of the population in Mehedinți County is a constant one. There is a steady trend of migration of the population in this area, which should be treated by various local measures and policies, in order to constantly preserve and develop this trend. We consider that the problem exists at the local level, because at the regional level, in Mehedinți County, there is another trend of population migration.

The econometric analysis highlights the differences between the analyzed localities from the perspectives of migration and of the labor market. In Mehedinți, there is a stronger correlation between the number of departures from the domicile and the average number of employees, suggesting that there is a tendency at the level of the population to leave domicile if the locality they belong to gathers large numbers of employees. Possible explanations for this result are the following:

- As more people get employed, less people find job opportunities in the same locality;
- As competitiveness grows in the local labor market, more people become interested in more developed or more attractive cities, even other countries;
- As there are less job opportunities in a locality, this constrains the local population to consider applying for jobs in different areas or maybe determines some to become entrepreneurs.

From the perspective of multicriteria analysis, depending on the importance of the indicator, Dolj County has obtained maximum values within indicators such as: the development of infrastructure based on the Danube River, the creation of new jobs through economic activities carried out in the vicinity of the Danube, the development of agritourism and the creation of prospects for accessing structural funds in rural areas, within the framework of measures specific to agricultural activities. The importance of these indicators at the level of this area is meant to develop local infrastructure on the help of the Danube River, so that the entrepreneurial environment can create new jobs in the framework of economic activities carried out in the vicinity of lakes, ponds, wetlands or the stretch of a small river. This will allow the development of agritourism naturally, and the use of rural structural funds on measures specific to agricultural activities will allow the maintenance of the level of the population in the rural area and possibly generate a wave of people coming from the urban to rural area.

## 5. Conclusions

As far as the current research is concerned, its purpose was to show the perspectives offered by the Mehedinti and Dolj Counties, the most overlooked areas of the Danube, in Romania, besides Tulcea County. This was done based on an econometric model and a multicriteria analysis. The complexity of the method used led to the presentation, naturally, of the high potential regarding the development of the entrepreneurial activity, an aspect that will allow the preservation and repopulation of the areas near the Danube. For example, in Mehedinti County, where population migration has a high level, and the variety of relief offers a broad perspective on the development of a business in the medium and long term, we can pursue the development of various economic activities, with the condition to offer an attractive salary package, which prevents a possible resignation of the employed staff. This aspect will allow, above all, the development of non-agricultural and recreational activities. On the other side, Dolj County has a stable workforce, an aspect that will allow, in an easier way, the development of agricultural and recreational activities. All these provide a unique purpose, developing the socio-economic perspective of the areas near the Danube, from the analyzed counties, so that the quality of life in these rural areas prospers, and those areas become more attractive for young generations.

The limits of this research are directly tied to characteristics of the data used in the cross-sectional linear regression model. More specifically, this analysis was focused on a single point in time to examine multiple subjects (the localities part of Mehedinți and Dolj Counties). Even though localities part of the Romanian Danube Region were analyzed and included in a viable econometric model, the reference year was 2018 and this implies that the evolution in time of the variables were not included in research. However, if the used data were to be a criteria based on which this research would be extended, then this study can be extended considering at least two points of view: from the perspective of the analyzed timeframe—the econometric model could be redesigned in order to include the evolution of the variables over a longer period in time and, from the perspective of the observations included in the econometric model, the beforementioned can be replicated on other localities part of the Danube Region. Not only that, but there are also other possibilities when it comes to extending this research. Our contribution resides in the framework we designed with the purpose of establishing a foundation for a sustainable action plan meant to increase the socio-economic attractivity of Dolj and Mehedinți Counties, at the level of each locality, part of the Romanian Danube Region. The novelty factor of this research paper refers to the way the multicriterial method was combined with an econometric method in order to highlight the current state and needs of development of the localities in the Romanian Danube Region. The sustainable action plan mentioned before should contain projects which can be

financed via the European Structural Funds or other financial instruments. Our research can help entrepreneurs willing to start new businesses in the Mehedinți and Dolj Counties by providing an in-depth analysis of the local labor market.

**Author Contributions:** Conceptualization, B.-C.C., M.C., M.-F.P. and A.S.; methodology, B.-C.C. and M.C.; software, B.-C.C. and M.C.; validation, B.-C.C., M.C., M.-F.P. and A.S.; formal analysis, M.-F.P., B.-C.C.; investigation, B.-C.C. and M.C.; resources, B.-C.C., M.C., M.-F.P. and A.S.; data curation, B.-C.C. and M.C.; writing—original draft preparation, B.-C.C., M.C., M.-F.P. and A.S.; writing—review and editing, M.-F.P., B.-C.C and M.C.; visualization, M.C.; supervision, B.-C.C. and M.-F.P.; project administration, B.-C.C. and M.-F.P.; funding acquisition, A.S. All authors have read and agreed to the published version of the manuscript.

**Funding:** The research leading to these results has been supported by the Romanian Ministry of Research and Innovation, through the Core Program (Program Nucleu)—"Developing integrated management for pilot areas of the Romanian Danube sector, influenced by climate change and anthropic interventions, by applying complex research methodologies," contract no. 13N/08.02.2019, carried out by NIRD GeoEcoMar. The financial support for this paper was provided by the Romanian Ministry of Research and Innovation, through the aforementioned project.

**Conflicts of Interest:** The authors declare no conflict of interest. The funders had no role in the design of the study; in the collection, analyses, or interpretation of data; in the writing of the manuscript, or in the decision to publish the results.

## Appendix A

**Table A1.** Calculation of the aggregated note.

| County ╲ The Indicator | I1 | I2 | I3 | I4 | I5 | I6 | I7 | I8 | I9 | I10 | TOTAL |
|---|---|---|---|---|---|---|---|---|---|---|---|
| Indicator's rank | 8 | 8 | 7 | 10 | 10 | 8 | 7 | 6 | 8 | 10 | - |
| Mehedinți County | 16 | 16 | 14 | 10 | 10 | 8 | 7 | 12 | 16 | 20 | 129 |
| Dolj County | 8 | 8 | 7 | 20 | 20 | 16 | 14 | 6 | 8 | 10 | 117 |

Source: authors' own conceptualization, based on the multicriteria analysis method.

## Appendix B

**Table A2.** The values of the analyzed indicators in the case of Dolj County, per locality, taking the year 2018 as the reference point.

| Item No. | Locality Code | Locality Name | POP107D | FOM104D | POP308A | FOM104 ÷ POP308A | POP308A ÷ POP308A |
|---|---|---|---|---|---|---|---|
| 1 | 70,520 | AFUMATI | 2587 | 105 | 53 | 4.06% | 2.05% |
| 2 | 70,566 | ALMAJ | 1867 | 126 | 39 | 6.75% | 2.09% |
| 3 | 70,637 | AMARASTII DE JOS | 5514 | 182 | 66 | 3.30% | 1.20% |
| 4 | 70,673 | AMARASTII DE SUS | 1637 | 57 | 26 | 3.48% | 1.59% |
| 5 | 70,726 | APELE VII | 2034 | 52 | 43 | 2.56% | 2.11% |
| 6 | 70,744 | ARGETOAIA | 4561 | 87 | 84 | 1.91% | 1.84% |
| 7 | 70,940 | BARCA | 4079 | 87 | 67 | 2.13% | 1.64% |
| 8 | 70,897 | BISTRET | 4224 | 310 | 83 | 7.34% | 1.96% |
| 9 | 70,968 | BOTOSESTI PAIA | 688 | 34 | 7 | 4.94% | 1.02% |
| 10 | 70,986 | BRABOVA | 1237 | 74 | 20 | 5.98% | 1.62% |
| 11 | 71,055 | BRADESTI | 4561 | 486 | 73 | 10.66% | 1.60% |
| 12 | 71,126 | BRALOSTITA | 3720 | 73 | 39 | 1.96% | 1.05% |
| 13 | 71,199 | BRATOVOESTI | 3200 | 158 | 53 | 4.94% | 1.66% |
| 14 | 71,260 | BREASTA | 4180 | 95 | 98 | 2.27% | 2.34% |
| 15 | 69,964 | BUCOVAT | 4190 | 461 | 70 | 11.00% | 1.67% |

**Table A2.** *Cont.*

| Item No. | Locality Code | Locality Name | POP107D | FOM104D | POP308A | FOM104 ÷ POP308A | POP308A ÷ POP308A |
|---|---|---|---|---|---|---|---|
| 16 | 71,340 | BULZESTI | 1374 | 55 | 17 | 4.00% | 1.24% |
| 17 | 71,607 | CALARASI | 5736 | 195 | 73 | 3.40% | 1.27% |
| 18 | 71,457 | CALOPAR | 3855 | 75 | 68 | 1.95% | 1.76% |
| 19 | 71,518 | CARAULA | 2496 | 68 | 53 | 2.72% | 2.12% |
| 20 | 74,859 | CARCEA | 2559 | 1566 | 57 | 61.20% | 2.23% |
| 21 | 74,867 | CARNA | 1353 | 41 | 31 | 3.03% | 2.29% |
| 22 | 71,536 | CARPEN | 2221 | 97 | 38 | 4.37% | 1.71% |
| 23 | 71,572 | CASTRANOVA | 3195 | 76 | 54 | 2.38% | 1.69% |
| 24 | 74,842 | CATANE | 2024 | 38 | 54 | 1.88% | 2.67% |
| 25 | 71,634 | CELARU | 4330 | 109 | 88 | 2.52% | 2.03% |
| 26 | 71,698 | CERAT | 4283 | 123 | 66 | 2.87% | 1.54% |
| 27 | 71,723 | CERNATESTI | 1745 | 55 | 31 | 3.15% | 1.78% |
| 28 | 71,787 | CETATE | 5372 | 142 | 99 | 2.64% | 1.84% |
| 29 | 71,812 | CIOROIASI | 1494 | 44 | 21 | 2.95% | 1.41% |
| 30 | 71,858 | CIUPERCENII NOI | 5167 | 86 | 62 | 1.66% | 1.20% |
| 31 | 71,885 | COSOVENI | 3276 | 254 | 57 | 7.75% | 1.74% |
| 32 | 71,910 | COTOFENII DIN DOS | 2275 | 92 | 37 | 4.04% | 1.63% |
| 33 | 74,875 | COTOFENII DIN FATA | 2012 | 80 | 37 | 3.98% | 1.84% |
| 34 | 71,956 | DANETI | 5747 | 95 | 89 | 1.65% | 1.55% |
| 35 | 72,034 | DESA | 4911 | 111 | 63 | 2.26% | 1.28% |
| 36 | 72,052 | DIOSTI | 2932 | 81 | 56 | 2.76% | 1.91% |
| 37 | 72,098 | DOBRESTI | 2354 | 94 | 52 | 3.99% | 2.21% |
| 38 | 74,883 | DOBROTESTI | 1637 | 40 | 34 | 2.44% | 2.08% |
| 39 | 72,150 | DRAGOTESTI | 2104 | 70 | 36 | 3.33% | 1.71% |
| 40 | 72,221 | DRANIC | 2326 | 82 | 41 | 3.53% | 1.76% |
| 41 | 72,276 | FARCAS | 2102 | 87 | 40 | 4.14% | 1.90% |
| 42 | 72,383 | GALICEA MARE | 4004 | 112 | 58 | 2.80% | 1.45% |
| 43 | 74,891 | GALICIUICA | 1379 | 31 | 14 | 2.25% | 1.02% |
| 44 | 72,579 | GANGIOVA | 2594 | 45 | 60 | 1.73% | 2.31% |
| 45 | 72,409 | GHERCESTI | 1691 | 748 | 32 | 44.23% | 1.89% |
| 46 | 74,907 | GHIDICI | 2434 | 56 | 51 | 2.30% | 2.10% |
| 47 | 74,915 | GHINDENI | 1832 | 38 | 18 | 2.07% | 0.98% |
| 48 | 72,463 | GIGHERA | 2802 | 55 | 44 | 1.96% | 1.57% |
| 49 | 72,506 | GIUBEGA | 2039 | 105 | 25 | 5.15% | 1.23% |
| 50 | 72,533 | GIURGITA | 2966 | 74 | 56 | 2.49% | 1.89% |
| 51 | 72,604 | GOGOSU | 558 | 27 | 10 | 4.84% | 1.79% |
| 52 | 72,640 | GOICEA | 2556 | 53 | 50 | 2.07% | 1.96% |
| 53 | 72,677 | GOIESTI | 3027 | 91 | 54 | 3.01% | 1.78% |
| 54 | 72,819 | GRECESTI | 1583 | 58 | 19 | 3.66% | 1.20% |
| 55 | 74,923 | INTORSURA | 1447 | 33 | 23 | 2.28% | 1.59% |
| 56 | 70,094 | ISALNITA | 4045 | 1601 | 52 | 39.58% | 1.29% |
| 57 | 72,882 | IZVOARE | 1512 | 48 | 18 | 3.17% | 1.19% |
| 58 | 72,926 | LEU | 4584 | 176 | 61 | 3.84% | 1.33% |
| 59 | 72,953 | LIPOVU | 3296 | 60 | 70 | 1.82% | 2.12% |
| 60 | 72,980 | MACESU DE JOS | 1255 | 58 | 18 | 4.62% | 1.43% |
| 61 | 73,013 | MACESU DE SUS | 1228 | 77 | 30 | 6.27% | 2.44% |
| 62 | 73,031 | MAGLAVIT | 4657 | 94 | 78 | 2.02% | 1.67% |
| 63 | 73,068 | MALU MARE | 5021 | 375 | 93 | 7.47% | 1.85% |
| 64 | 73,317 | MARSANI | 4527 | 107 | 67 | 2.36% | 1.48% |
| 65 | 73,102 | MELINESTI | 3882 | 247 | 80 | 6.36% | 2.06% |
| 66 | 73,246 | MISCHII | 1637 | 60 | 34 | 3.67% | 2.08% |
| 67 | 73,335 | MOTATEI | 6941 | 156 | 93 | 2.25% | 1.34% |
| 68 | 73,371 | MURGASI | 2328 | 55 | 46 | 2.36% | 1.98% |
| 69 | 73,460 | NEGOI | 2400 | 54 | 48 | 2.25% | 2.00% |
| 70 | 73,503 | ORODEL | 2538 | 61 | 58 | 2.40% | 2.29% |
| 71 | 73,567 | OSTROVENI | 4959 | 98 | 84 | 1.98% | 1.69% |
| 72 | 73,594 | PERISOR | 1666 | 78 | 36 | 4.68% | 2.16% |
| 73 | 73,629 | PIELESTI | 3763 | 505 | 93 | 13.42% | 2.47% |
| 74 | 73,665 | PISCU VECHI | 2702 | 89 | 34 | 3.29% | 1.26% |
| 75 | 73,709 | PLENITA | 4533 | 274 | 91 | 6.04% | 2.01% |

**Table A2.** *Cont.*

| Item No. | Locality Code | Locality Name | POP107D | FOM104D | POP308A | FOM104 ÷ POP308A | POP308A ÷ POP308A |
|---|---|---|---|---|---|---|---|
| 76 | 74,931 | PLESOI | 1322 | 40 | 33 | 3.03% | 2.50% |
| 77 | 70,110 | PODARI | 6753 | 829 | 130 | 12.28% | 1.93% |
| 78 | 73,736 | POIANA MARE | 10445 | 645 | 143 | 6.18% | 1.37% |
| 79 | 73,772 | PREDESTI | 2061 | 54 | 42 | 2.62% | 2.04% |
| 80 | 73,852 | RADOVAN | 1362 | 77 | 31 | 5.65% | 2.28% |
| 81 | 73,905 | RAST | 3585 | 76 | 54 | 2.12% | 1.51% |
| 82 | 73,923 | ROBANESTI | 2299 | 122 | 33 | 5.31% | 1.44% |
| 83 | 74,949 | ROJISTE | 2470 | 52 | 53 | 2.11% | 2.15% |
| 84 | 73,996 | SADOVA | 8760 | 166 | 96 | 1.89% | 1.10% |
| 85 | 74,028 | SALCUTA | 2176 | 46 | 39 | 2.11% | 1.79% |
| 86 | 74,073 | SCAESTI | 2087 | 57 | 35 | 2.73% | 1.68% |
| 87 | 74,108 | SEACA DE CAMP | 1797 | 46 | 21 | 2.56% | 1.17% |
| 88 | 74,135 | SEACA DE PADURE | 918 | 48 | 24 | 5.23% | 2.61% |
| 89 | 74,171 | SECU | 1047 | 37 | 23 | 3.53% | 2.20% |
| 90 | 74,224 | SILISTEA CRUCII | 1487 | 50 | 25 | 3.36% | 1.68% |
| 91 | 70,174 | SIMNICU DE SUS | 4857 | 260 | 95 | 5.35% | 1.96% |
| 92 | 74,242 | SOPOT | 1709 | 56 | 27 | 3.28% | 1.58% |
| 93 | 74,956 | TALPAS | 1243 | 33 | 27 | 2.65% | 2.17% |
| 94 | 74,322 | TEASC | 3030 | 90 | 53 | 2.97% | 1.75% |
| 95 | 74,359 | TERPEZITA | 1519 | 112 | 29 | 7.37% | 1.91% |
| 96 | 74,411 | TESLUI | 2228 | 60 | 36 | 2.69% | 1.62% |
| 97 | 74,509 | TUGLUI | 2906 | 97 | 45 | 3.34% | 1.55% |
| 98 | 74,536 | UNIREA | 3762 | 61 | 53 | 1.62% | 1.41% |
| 99 | 74,554 | URZICUTA | 3018 | 68 | 64 | 2.25% | 2.12% |
| 100 | 74,581 | VALEA STANCIULUI | 5295 | 153 | 97 | 2.89% | 1.83% |
| 101 | 74,732 | VARTOP | 1712 | 75 | 28 | 4.38% | 1.64% |
| 102 | 74,750 | VARVORU DE JOS | 2483 | 67 | 41 | 2.70% | 1.65% |
| 103 | 74,616 | VELA | 1829 | 52 | 23 | 2.84% | 1.26% |
| 104 | 74,705 | VERBITA | 1279 | 45 | 24 | 3.52% | 1.88% |

Source: authors' own processing (data source: The Romanian National Institute of Statistics).

**Table A3.** The values of the analyzed indicators in the case of Mehedinţi County, per locality, taking the year 2018 as the reference point.

| Item No. | Locality Code | Locality Name | POP107D | FOM104D | POP308A | FOM104 ÷ POP308A | POP308A ÷ POP308A |
|---|---|---|---|---|---|---|---|
| 1 | 110,571 | BACLES | 1825 | 98 | 38 | 5.37% | 2.08% |
| 2 | 110,296 | BALA | 3692 | 178 | 108 | 4.82% | 2.93% |
| 3 | 110,535 | BALACITA | 2666 | 92 | 57 | 3.45% | 2.14% |
| 4 | 110,456 | BALTA | 1021 | 43 | 38 | 4.21% | 3.72% |
| 5 | 110,688 | BALVANESTI | 918 | 50 | 14 | 5.45% | 1.53% |
| 6 | 114,060 | BRANISTEA | 1830 | 53 | 40 | 2.90% | 2.19% |
| 7 | 110,740 | BREZNITA-MOTRU | 1386 | 81 | 38 | 5.84% | 2.74% |
| 8 | 110,820 | BREZNITA-OCOL | 3976 | 288 | 57 | 7.24% | 1.43% |
| 9 | 110,875 | BROSTENI | 2762 | 164 | 52 | 5.94% | 1.88% |
| 10 | 110,946 | BURILA MARE | 2027 | 85 | 42 | 4.19% | 2.07% |
| 11 | 111,006 | BUTOIESTI | 3217 | 142 | 65 | 4.41% | 2.02% |
| 12 | 111,097 | CAZANESTI | 2077 | 74 | 53 | 3.56% | 2.55% |
| 13 | 111,220 | CIRESU | 481 | 65 | 13 | 13.51% | 2.70% |
| 14 | 111,275 | CORCOVA | 6001 | 254 | 105 | 4.23% | 1.75% |
| 15 | 111,417 | CORLATEL | 1281 | 49 | 15 | 3.83% | 1.17% |
| 16 | 111,444 | CUJMIR | 3298 | 272 | 59 | 8.25% | 1.79% |
| 17 | 111,550 | DARVARI | 2557 | 94 | 47 | 3.68% | 1.84% |
| 18 | 111,480 | DEVESEL | 3011 | 126 | 74 | 4.18% | 2.46% |
| 19 | 112,904 | DUBOVA | 940 | 76 | 17 | 8.09% | 1.81% |

**Table A3.** *Cont.*

| Item No. | Locality Code | Locality Name | POP107D | FOM104D | POP308A | FOM104 ÷ POP308A | POP308A ÷ POP308A |
|---|---|---|---|---|---|---|---|
| 20 | 111,587 | DUMBRAVA | 1351 | 40 | 23 | 2.96% | 1.70% |
| 21 | 112,245 | ESELNITA | 2894 | 233 | 53 | 8.05% | 1.83% |
| 22 | 111,685 | FLORESTI | 2570 | 75 | 61 | 2.92% | 2.37% |
| 23 | 111,783 | GARLA MARE | 3646 | 142 | 59 | 3.89% | 1.62% |
| 24 | 111,818 | GODEANU | 555 | 24 | 23 | 4.32% | 4.14% |
| 25 | 111,863 | GOGOSU | 4298 | 350 | 98 | 8.14% | 2.28% |
| 26 | 111,916 | GRECI | 1202 | 56 | 19 | 4.66% | 1.58% |
| 27 | 111,989 | GROZESTI | 2007 | 45 | 39 | 2.24% | 1.94% |
| 28 | 112,030 | GRUIA | 3031 | 76 | 70 | 2.51% | 2.31% |
| 29 | 112,076 | HINOVA | 2897 | 225 | 48 | 7.77% | 1.66% |
| 30 | 112,129 | HUSNICIOARA | 1226 | 134 | 34 | 10.93% | 2.77% |
| 31 | 112,263 | ILOVAT | 1197 | 77 | 28 | 6.43% | 2.34% |
| 32 | 112,334 | ILOVITA | 1304 | 73 | 31 | 5.60% | 2.38% |
| 33 | 112,370 | ISVERNA | 2087 | 122 | 43 | 5.85% | 2.06% |
| 34 | 112,469 | IZVORU BARZII | 2761 | 400 | 55 | 14.49% | 1.99% |
| 35 | 112,548 | JIANA | 4611 | 199 | 85 | 4.32% | 1.84% |
| 36 | 112,600 | LIVEZILE | 1417 | 72 | 33 | 5.08% | 2.33% |
| 37 | 112,664 | MALOVAT | 2588 | 225 | 65 | 8.69% | 2.51% |
| 38 | 112,744 | OBARSIA DE CAMP | 1727 | 47 | 34 | 2.72% | 1.97% |
| 39 | 110,027 | OBARSIA-CLOSANI | 1023 | 78 | 37 | 7.62% | 3.62% |
| 40 | 112,771 | OPRISOR | 2133 | 142 | 64 | 6.66% | 3.00% |
| 41 | 112,806 | PADINA | 1199 | 39 | 32 | 3.25% | 2.67% |
| 42 | 112,879 | PATULELE | 3660 | 139 | 67 | 3.80% | 1.83% |
| 43 | 112,959 | PODENI | 825 | 47 | 11 | 5.70% | 1.33% |
| 44 | 112,995 | PONOARELE | 2397 | 205 | 63 | 8.55% | 2.63% |
| 45 | 113,153 | POROINA MARE | 929 | 54 | 28 | 5.81% | 3.01% |
| 46 | 113,206 | PRISTOL | 1384 | 54 | 28 | 3.90% | 2.02% |
| 47 | 113,233 | PRUNISOR | 1877 | 114 | 40 | 6.07% | 2.13% |
| 48 | 113,395 | PUNGHINA | 3256 | 81 | 100 | 2.49% | 3.07% |
| 49 | 113,466 | ROGOVA | 1434 | 69 | 52 | 4.81% | 3.63% |
| 50 | 113,493 | SALCIA | 2716 | 110 | 48 | 4.05% | 1.77% |
| 51 | 109,826 | SIMIAN | 10,346 | 1397 | 221 | 13.50% | 2.14% |
| 52 | 113,625 | SISESTI | 2568 | 119 | 53 | 4.63% | 2.06% |
| 53 | 113,698 | SOVARNA | 1081 | 97 | 22 | 8.97% | 2.04% |
| 54 | 113,518 | STANGACEAUA | 1262 | 41 | 44 | 3.25% | 3.49% |
| 55 | 113,607 | SVINITA | 931 | 53 | 24 | 5.69% | 2.58% |
| 56 | 113,732 | TAMNA | 3294 | 98 | 80 | 2.98% | 2.43% |
| 57 | 113,849 | VANATORI | 1929 | 60 | 41 | 3.11% | 2.13% |
| 58 | 113,894 | VANJULET | 1893 | 90 | 31 | 4.75% | 1.64% |
| 59 | 113,929 | VLADAIA | 1592 | 73 | 23 | 4.59% | 1.44% |
| 60 | 113,974 | VOLOIAC | 1668 | 52 | 30 | 3.12% | 1.80% |
| 61 | 114,079 | VRATA | 1969 | 58 | 27 | 2.95% | 1.37% |

Source: authors' own processing (data source: The Romanian National Institute of Statistics).

## Appendix C

| obs | Actual | Fitted | Residual |
|---|---|---|---|
| 110571 BACLES | 38.0000 | 44.0409 | -6.04090 |
| 110296 BALA | 108.000 | 55.4514 | 52.5486 |
| 110535 BALACITA | 57.0000 | 43.1851 | 13.8149 |
| 110456 BALTA | 38.0000 | 36.1962 | 1.80381 |
| 110688 BALVANESTI | 14.0000 | 37.1946 | -23.1946 |
| 114060 BRANISTEA | 40.0000 | 37.6225 | 2.37750 |
| 110740 BREZNITA-MOTRU | 38.0000 | 41.6162 | -3.61617 |
| 110820 BREZNITA-OCOL | 57.0000 | 71.1408 | -14.1408 |
| 110875 BROSTENI | 52.0000 | 53.4546 | -1.45456 |
| 110946 BURILA MARE | 42.0000 | 42.1867 | -0.18670 |
| 111006 BUTOIESTI | 65.0000 | 50.3167 | 14.6833 |
| 111097 CAZANESTI | 53.0000 | 40.6178 | 12.3822 |
| 111220 CIRESU | 13.0000 | 39.3341 | -26.3341 |
| 111275 CORCOVA | 105.000 | 66.2914 | 38.7086 |
| 111417 CORLATEL | 15.0000 | 37.0520 | -22.0520 |
| 111444 CUJMIR | 59.0000 | 68.8587 | -9.85873 |
| 111550 DARVARI | 47.0000 | 43.4704 | 3.52962 |
| 111480 DEVESEL | 74.0000 | 48.0346 | 25.9654 |
| 112904 DUBOVA | 17.0000 | 40.9030 | -23.9030 |
| 111587 DUMBRAVA | 23.0000 | 35.7683 | -12.7683 |
| 112245 ESELNITA | 53.0000 | 63.2961 | -10.2961 |
| 111685 FLORESTI | 61.0000 | 40.7604 | 20.2396 |
| 111783 GARLA MARE | 59.0000 | 50.3167 | 8.68332 |
| 111818 GODEANU | 23.0000 | 33.4862 | -10.4862 |
| 111863 GOGOSU | 98.0000 | 79.9840 | 18.0160 |
| 111916 GRECI | 19.0000 | 38.0504 | -19.0504 |
| 111989 GROZESTI | 39.0000 | 36.4814 | 2.51855 |
| 112030 GRUIA | 70.0000 | 40.9030 | 29.0970 |
| 112076 HINOVA | 48.0000 | 62.1551 | -14.1551 |
| 112129 HUSNICIOARA | 34.0000 | 49.1756 | -15.1756 |
| 112263 ILOVAT | 28.0000 | 41.0456 | -13.0456 |
| 112334 ILOVITA | 31.0000 | 40.4751 | -9.47512 |
| 112370 ISVERNA | 43.0000 | 47.4641 | -4.46405 |
| 112469 IZVORU BARZII | 55.0000 | 87.1155 | -32.1155 |
| 112548 JIANA | 85.0000 | 58.4467 | 26.5533 |
| 112600 LIVEZILE | 33.0000 | 40.3325 | -7.33249 |
| 112664 MALOVAT | 65.0000 | 62.1551 | 2.84493 |
| 112744 OBARSIA DE CAMP | 34.0000 | 36.7667 | -2.76671 |
| 110027 OBARSIA-CLOSANI | 37.0000 | 41.1883 | -4.18828 |
| 112771 OPRISOR | 64.0000 | 50.3167 | 13.6833 |
| 112806 PADINA | 32.0000 | 35.6257 | -3.62566 |
| 112879 PATULELE | 67.0000 | 49.8888 | 17.1112 |
| 112959 PODENI | 11.0000 | 36.7667 | -25.7667 |
| 112995 PONOARELE | 63.0000 | 59.3024 | 3.69756 |
| 113153 POROINA MARE | 28.0000 | 37.7651 | -9.76513 |
| 113206 PRISTOL | 28.0000 | 37.7651 | -9.76513 |
| 113233 PRUNISOR | 40.0000 | 46.3230 | -6.32300 |
| 113395 PUNGHINA | 100.000 | 41.6162 | 58.3838 |
| 113466 ROGOVA | 52.0000 | 39.9046 | 12.0954 |
| 113493 SALCIA | 48.0000 | 45.7525 | 2.24752 |
| 109826 SIMIAN | 221.000 | 229.319 | -8.31885 |
| 113625 SISESTI | 53.0000 | 47.0362 | 5.96384 |
| 113698 SOVARNA | 22.0000 | 43.8983 | -21.8983 |
| 113518 STANGACEAUA | 44.0000 | 35.9109 | 8.08908 |
| 113607 SVINITA | 24.0000 | 37.6225 | -13.6225 |
| 113732 TAMNA | 80.0000 | 44.0409 | 35.9591 |
| 113849 VANATORI | 41.0000 | 38.6209 | 2.37908 |
| 113894 VANJULET | 31.0000 | 42.8999 | -11.8994 |
| 113929 VLADAIA | 23.0000 | 40.4751 | -17.4751 |
| 113974 VOLOIAC | 30.0000 | 37.4799 | -7.47987 |
| 114079 VRATA | 27.0000 | 38.3357 | -11.3357 |

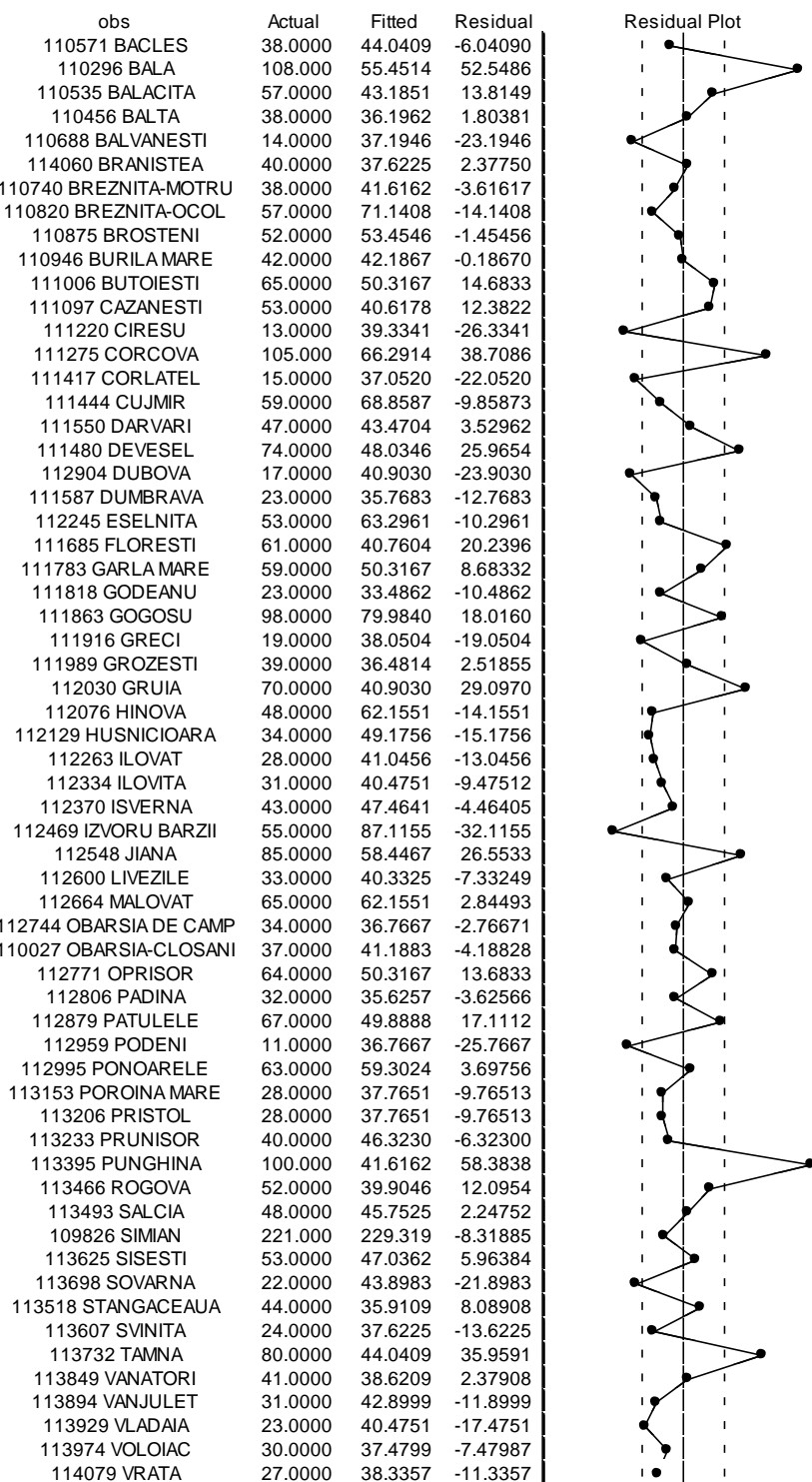

**Figure A1.** The residual plot of the model with the observations. Actual and fitted residuals. Reference point: the year 2018. Observations: the localities part of Mehedinți County. Source: authors' own representation (data source: The Romanian National Institute of Statistics, processed in EViews 10 Student Version Lite, IHS Global Inc., Irvine, CA, USA).

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
