# Peer review of "The Socio-Economic Impact of Migration on the Labor Market in the Romanian Danube Region"

_sustainability, doi:10.3390/su12208654_

Round 1

Reviewer 1 Report

This is a very good and interesting paper, regarding an important research topic. Various research methods and tools were applied, the structure is clear and coherent. I recommend for publication.

Author Response

Thank you so much for your feedback and for your patience to read and review our article! We are truly grateful for your interest in our article!

Reviewer 2 Report

This is an interesting paper on migration flows and employment in Romania, using regressions and a qualitative analysis. The findings indicate differing levels of migration relative to employment across the Romanian regions. I have the following comments:

  • In the motivation for choosing the two regions, the authors say because ' they represent the most passed area of the Danube river'. (line 141). This needs more clarification.
  • In Table 1. there are some importance coefficients, but I am not sure where these values have come from, this needs to be explained especially the criteria used to produce them. Similarly with Table 2.
  • The data refers to the number of departures from the area, as well as the number of employees. Wouldn't these factors though depend on the populations in these areas. Is there data for the total population and if there is could this be used to create ratios of these variables? This point needs at least more explanation.
  • Some theoretical discussion needs to be added for the regression model, is it based on a common theoretical model?
  • In the tables of results, you only need to include the main information that is discussed in the text, not all the computer output.
  • Overall, the link between the multicriterial analysis and the regression analysis needs to be made clearer, especially in the introduction. At the moment they seem to be two separate sections.

Author Response

Point 1: In the motivation for choosing the two regions, the authors say because ' they represent the most passed area of the Danube river'. (line 141). This needs more clarification.
Response 1: We modified the wording between lines 145-147, so that the information provided is more explicit.

Point 2: In Table 1. there are some important coefficients, but I am not sure where these values have come from, this needs to be explained especially the criteria used to produce them. Similarly with Table 2.
Response 2: Between rows 196-201, it was explained how to assign the values of the coefficients chosen for analysis in the first table, while between rows 312-318 was mentioned the purpose of the link between tables 1 and 2, according to the method of multicriteria analysis (more precisely, the determination of the global indicator - the formula that was corrected in this new format of the paper, where k = 2). Also, an adaptation of the previous text was made in order to achieve a harmonious combination with the new specified information.

Point 3: The data refers to the number of departures from the area, as well as the number of employees. Wouldn't these factors though depend on the populations in these areas. Is there data for the total population and if there is could this be used to create ratios of these variables? This point needs at least more explanation.
Response 3: Data related to the variables and to the total population, including the variable ratios were included and discussed in the body of the paper. The values of the indicators were added in the Appendix. (Lines: 287-292, 440-456, 660-666)

Point 4: Some theoretical discussion needs to be added for the regression model, is it based on a common theoretical model?
Response 4: We added further theoretical discussions regarding the regression model. (Lines: 260-263 and 472-473)

Point 5: In the tables of results, you only need to include the main information that is discussed in the text, not all the computer output.
Response 5: We consider all the regression output necessary, even though some indicators were not approached in detail. If we would cut some of the regression output, then this could lead to confusion regarding the method we used. It is a traditional quantitative research method in econometrics and the regression output within the paper is common in other publications.

Point 6: Overall, the link between the multicriterial analysis and the regression analysis needs to be made clearer, especially in the introduction. At the moment they seem to be two separate sections.
Response 6: We have modified the title of the paper in "The Socio-Economic Impact of Migration on the Labor Market in the Romanian Danube Region". Moreover, we have added lines 133-138 in the Introduction section to made things more clear and straightforward. 

Thank you so much for all your observations!

Reviewer 3 Report

There are few aspects to be clarified:

1- the title includes ”Impact of Migration Flows” - I didn't find in the paper any references about migration flows. The paper is not correlated with the title

2- Romanian Danube region is related with 12 counties - Why were analyzed only 2 of them, why Mehedinti and Dolj? Are this counties the most relevant for the labor market, migration flows or socio-economic context?

3- The table 1 - Coefficient of importance of the influencing factors. How was the value of this coefficients determined?

4- Table 2 - It looks that only 16 and 8 are the values for the indicators - this doesn't look valid and real

5- If the main conclusion that Mehedinti has a greater potential than Dolj based on the scores value (Table 2 - pg 7) What is the reason for the deep analysis? 

6- I do not see the relevance of the paper for the specialists and I consider that the interest of the readers is low.

7- The references list is large but not very proper. Half of it is based on Romanian authors. Not relevant references about labor market or similar studies about determining the labor potential.

8- reference 4 - is not cited proper.

Author Response

Point 1 - the title includes ”Impact of Migration Flows” - I didn't find in the paper any references about migration flows. The paper is not correlated with the title
Response 1 - We have modified the title of the paper in "The Socio-Economic Impact of Migration on the Labor Market in the Romanian Danube Region". We have eliminated the "migration flows" from the paper, and kept only "migration".

Point 2 - Romanian Danube region is related with 12 counties - Why were analyzed only 2 of them, why Mehedinti and Dolj? Are this counties the most relevant for the labor market, migration flows or socio-economic context?
Response 2 - The two counties were selected for the analysis of labor migration, because in this region the Danube reaches the surface of the shores in most places, at national level, except for Tulcea County, where the Danube Delta is formed. The other counties in Romania crossed by the Danube report a lower share regarding the wetlands of the shores (according to lines 145-147). The two counties do not have an important relevance regarding the labor market and socio-economic context at national level, but the research aims to identify the existing situation on the wettest area in Romania, crossed by the Danube, except Tulcea County, where the Danube Delta is formed. There is a lot of research on the Danube Delta, but our proposal to approach the two counties in the southwest of the country, offers the paper a uniqueness in the field of economic research.

Point 3 - The table 1 - Coefficient of importance of the influencing factors. How was the value of this coefficients determined?
Response 3 - Between rows 196-201, it was explained how to assign the values of the coefficients chosen for analysis in the first table.

Point 4 - Table 2 - It looks that only 16 and 8 are the values for the indicators - this doesn't look valid and real
Response 4 - Indeed, there was an error in determining the global indicator. We modified the values in Table no.2 and Appendix A. Thank you so much for this observation!

Point 5 - If the main conclusion that Mehedinti has a greater potential than Dolj based on the scores value (Table 2 - pg 7) What is the reason for the deep analysis? 
Response 5 - We have added lines 133-138 "Starting with a multicriteria analysis meant to evaluate important characteristics of Mehedinţi and Dolj Counties, this study goes deeper into the labor market analysis, from the county level analysis (multicriteria method) to the localities level analysis (econometric method, based on the initial findings of the multicriteria analysis), aiming to offer a deep analysis of the socio-economic potential of the area and pursuing the provision of various solutions aimed for the development of the area." Moreover, throughout the paper, we have treated with equal importance the two counties, and in the Conclusion section, we have highlighted economic issues and offered solutions for both counties. (lines 610-619)

Point 6 - I do not see the relevance of the paper for the specialists and I consider that the interest of the readers is low.
Response 6The aim of the paper was to analyze the socio-economic impact of migration on the labor force in the wettest areas, crossed by the Danube, except for Tulcea County (where the Danube Delta is formed). The choice of theme is due to the fact that this region of Romania has a high potential for the development and support of entrepreneurship through economic activities that can be achieved through the use of Danube waterways. As each region has its specificity, the southwestern part of Romania enjoys the presence of the Danube, an aspect that must be capitalized and promoted at national and even global level. Moreover, the relevance of this paper resides in the fact that we designed a framework with the purpose of establishing the foundation for a sustainable action plan meant to increase the socio-economic attractivity of Dolj and Mehedinţi Counties, at the level of each locality part of the Romanian Danube Region. The novelty factor of this research paper refers to the way the multicriterial method was combined with an econometric method in order to highlight the current state and needs of the development of the localities in the Romanian Danube Region. The sustainable action plan mentioned before should contain projects which can be financed via European Structural Funds or any other financial instruments. Our research can help entrepreneurs willing to start new businesses in the Mehedinţi and Dolj Counties by providing an in-depth analysis of the local workforce.

Point 7 - The references list is large but not very proper. Half of it is based on Romanian authors. Not relevant references about labor market or similar studies about determining the labor potential.
Response 7 - Our work refers only to Romania and then it is normal to have a large number of references of Romanian authors, because they wrote works about specific areas in Romania. Though we did emphasize in the paper international papers and we did expand our reference list with #7, #10, #11 and #52.

Point 8 - reference 4 - is not cited proper.
Response 8 - The error regarding citing reference #4 has been corrected (the name was changed with the surname), as can be seen now in the reference list.

Thank you so much for all your observations!

Round 2

Reviewer 2 Report

The points raised earlier have been addressed well, although could you just clarify one point.  I am not sure if the data in ratio form was used for the regression analysis, or was it only used for the overall statistics as in table B1? If it was only the overall statistics, it would also need to be used for the regressions.

Author Response

The data in ratio was added (Appendix B) and discussed (Lines: 287-292, 440-456, 660-666). Methodologically, it is not recommended to add these ratios into the regression model because it would cause spurious results (Tu et al., 2004). To be more precise, in our case, the coefficient of determination would grow spuriously from 66.17 to 87.04%, because the indicators are already analyzed from the perspective of the raw data and adding the ratios of the raw data to the model would only add spurious "explanations" to the dependent variable. Moreover, adding these ratios to the model would invalidate it from the perspective of the White Test regarding the residuals: they would become heteroscedastic, considering the Prob.F 0.00 and Prob. χ2 0.00.

For more details regarding spurious results with ratio variables, please see:
Tu, Yu-Kang, Valerie Clerehugh, and Mark S. Gilthorpe. "Ratio variables in regression analysis can give rise to spurious results: illustration from two studies in periodontology." Journal of dentistry 32.2 (2004): 143-151.
Phillips, Peter CB. "Understanding spurious regressions in econometrics." Journal of econometrics 33.3 (1986): 311-340.

Reviewer 3 Report

The paper gives more details now, that makes it satisfactory.

Author Response

Thank you very much for all your valuable observations!